# Anti-Inflammatory and Hypouricemic Effect of Bioactive Compounds: Molecular Evidence and Potential Application in the Management of Gout

Anna Scanu [1], Roberto Luisetto [2], Roberta Ramonda [1], Paolo Spinella [3], Paolo Sfriso [1], Paola Galozzi [1] and Francesca Oliviero [1,*]

1   Rheumatology Unit, Department of Medicine—DIMED, University of Padova, 35128 Padova, Italy
2   Department of Surgery, Oncology and Gastroenterology—DISCOG, University of Padova, 35128 Padova, Italy
3   Clinical Nutrition Unit, Department of Medicine—DIMED, University of Padova, 35128 Padova, Italy
*   Correspondence: francesca.oliviero@unipd.it

**Abstract:** Gout is caused by the deposition of monosodium urate crystals in the joint and represents the most common form of inflammatory arthritis in men. Its prevalence is rising worldwide mainly due to the increase of risk factors associated with the disease, in particular hyperuricemia. Besides gout, hyperuricemia leads to an increased inflammatory state of the body with consequent increased risk of comorbidities such as cardiovascular diseases. Increasing evidence shows that bioactive compounds have a significant role in fighting inflammatory and immune chronic conditions. In gout and hyperuricemia, these molecules can exert their effects at two levels. They can either decrease serum uric acid concentrations or fight inflammation associated with monosodium urate crystals deposits and hyperuricemia. In this view, they might be considered valuable support to the pharmacological therapy and prevention of the disease. This review aims to provide an overview of the beneficial role of bioactive compounds in hyperuricemia, gout development, and inflammatory pathways of the disease.

**Keywords:** bioactive compounds; gout; hyperuricemia; inflammation; crystal-induced arthritis

## 1. Introduction

Over the last decades, bioactive compounds have received considerable attention for their anti-inflammatory, antioxidant, anti-tumoral, and immunomodulating properties. Flavonoids, phenolic acids, alkaloids, saponins, and polysaccharides, derived from fruits and vegetables, are only some of the molecules showing a range of beneficial effects in different experimental models of inflammatory diseases.

Gout is the most common form of inflammatory arthritis in men, with a strong impact on individual health and healthcare systems. In developed countries, the prevalence of gout is 3–6% in males and 1–2% in females but is expected to rise worldwide mainly due to the increased exposure to behavioral, environmental, and metabolic risk factors associated with the disease, in particular hyperuricemia [1,2].

This review aims to provide an overview of the beneficial role of bioactive compounds in gout pathogenesis and progression. It examines how these compounds modulate uric acid levels and the inflammatory pathways in this disease.

## 2. Gout and Hyperuricemia

### 2.1. The Inflammatory Process in Gout

Gout is caused by the deposition of monosodium urate (MSU) crystals in articular and periarticular tissues. These crystals trigger an acute and painful inflammatory reaction with the recruitment of inflammatory cells, mainly polymorphonucleate neutrophils, at the

site of inflammation with a consequent increase in synovial fluid volume and limited joint function [3].

The principal mechanism involved in crystal-induced inflammation is the activation of the cytoplasmatic NACHT-LRRPYD-containing protein-3 (NLRP) 3 inflammasome with a sustained release of IL-1ß. This is a two steps process encompassing cell priming and activation. A first non-specific signal activates innate immune receptors inducing the expression of the precursor proteins that are the substrates of inflammatory caspases. A second signal, more specific, is provided by the interaction of MSU crystals with cells and leads to inflammasome assembly and activation [4]. Of interest, a role of glucose metabolism during crystal-induced inflammation has been recently described in primed THP1 cells, suggesting that metabolic changes in macrophages after MSU crystals activation are relevant in gout [5].

If not treated, gout leads to severe damage of articular tissues and subchondral bones with consequent disability. Furthermore, the persistent increase in serum levels of uric acid leads to comorbidities, in particular those related to cardiovascular diseases [6].

### 2.2. Hyperuricemia-Associated Inflammation

Hyperuricemia is the main risk factor for gout and the prerequisite for crystal precipitation and the development of the disease. Hyperuricemia occurs when plasma uric acid exceeds 0.36 mmol/L (6 mg/dL) [3]. It is associated with different conditions which lead to either increased metabolic production or a decreased renal excretion of uric acid (Figure 1).

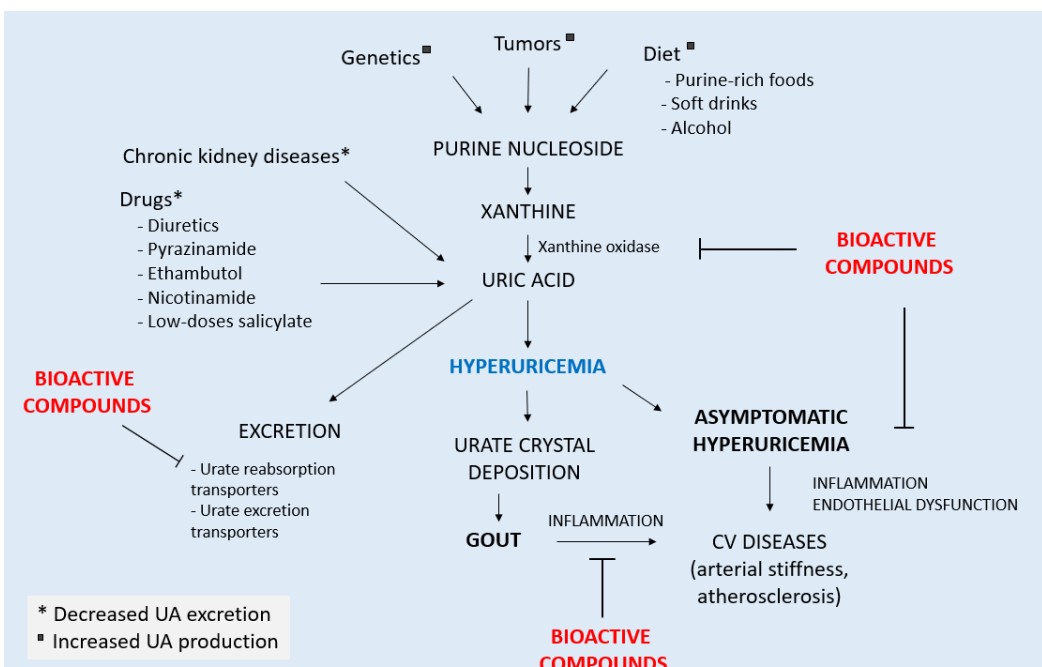

**Figure 1.** Conditions leading to hyperuricemia and gout in humans.

In the absence of episodes of inflammation triggered by MSU crystals, hyperuricemia is referred to as asymptomatic. Hyperuricemia has been recognized as an independent risk factor for mortality in the general population. Indeed, data show that elevated levels of uric acid contribute to inflammation, oxidative stress, and endothelial dysfunction increases the risk of developing cardiovascular and chronic kidney diseases [7].

Uric acid has been shown to sustain the atherosclerosis process via distressing lipid metabolism, reducing the synthesis of nitric oxide in endothelial cells, and promoting the proliferation of vascular smooth muscle cells [8].

During the asymptomatic phase between gouty flares, called intercritical gout, it has been observed that uric acid causes an unbalance between the transcription levels of IL-1ß and its natural inhibitor IL-1Ra in human monocyte cells. This effect amplifies the reactivity of cells to inflammatory stimuli and is epigenetically mediated by histone methylation [9].

As a consequence, all hyperuricemic individuals could be at risk of an increased state of inflammation and associated conditions.

Uric acid is the final product of purine nucleoside metabolism. Genetic diseases, tumors, and diet can lead to increased purine catabolism and uric acid production. Chronic kidney disease and certain drugs cause a decrease in renal excretion of uric acid increasing its circulating levels. Hyperuricemia is a risk factor for gout development and cardiovascular diseases (Figure 1).

## 3. Beneficial Effect of Bioactive Compounds in Gout and Hyperuricemia

Bioactive compounds are defined as essential and non-essential compounds that occur in nature, are part of the food chain, and can be shown to affect human health [10].

A growing body of evidence supports a role for these molecules in both diminishing circulating uric acid levels and fighting inflammation associated with MSU crystals deposits and hyperuricemia (Figure 1).

### 3.1. Hypouricemic Properties of Bioactive Compounds

A remarkable number of bioactive compounds have been shown to possess hypouricemic effects and are considered in the prevention and management of hyperuricemia [11]. The decrease of uric acid levels occurs mainly through the inhibition of xanthine oxidase (XOD) and adenosine deaminase (ADA), enzymes that regulate the metabolic pathway leading to uric acid synthesis (Figure 2). In particular, the ability to inhibit XOD has been investigated for a large number of molecules including caffeic acid [12], apigenin, quercetin, luteolin [13,14], kaempferol [15], rutin [16], chlorogenic acid [17], sinapic acid [18], phloroglucinol derivatives [19], and polyphenol-rich plant extracts [20,21]. Among the latter, extracts from tea leaves, Tartary buckwheat, Chrysanthemum morifolium, Smilax china L., Flourensia fiebrigii, and Keladi Candik, that contain a mix of many different bioactive compounds, have shown a high XOD inhibitory activity [22–26]. Molecular docking studies have demonstrated that these compounds can interact with some XOD amino acid residues, forming a complex which effectively interferes with the enzyme activity [11,15,27]. Phenolic compounds with XOD inhibitory and antioxidant activity have been extracted not only from plants but also from animals or their products, such as Pieris brassicacea larvae and cow milk [28,29].

Of interest, the hypouricemic effect of phenolic and other bioactive compounds is often associated with antioxidant and anti-inflammatory activity [16,17]. Boswellia dalziellii extracts showed inhibitory activity against 5-Lipoxygenase (5-LO), which is the enzyme responsible for the biosynthesis of leukotrienes, a group of lipid mediators of inflammation derived from arachidonic acid [30]. Morin and other constituents of Maclura cochinchinensis revealed downregulation of TNF-α, TGF-β, iNOS, and COX-2 mRNA expression in RAW 264.7 mouse macrophage cell lines stimulated with lipopolysaccharide (LPS) [31]. Extract of the fungus Phellinus igniarius, in addition to an XOD inhibitory activity, also exhibited a decrease in the release of IL-1β and ICAM-1 using an in vitro model of RAW 264.7 cells stimulated with a sodium urate solution [32].

Recently, the ability of apple polyphenol to inhibit XOD has been demonstrated using a cell-based assay evaluating intracellular oxidase activity. This sensitive method which assesses the effect of agents able to cross the cell membrane is more predictive of human response [33].

Finally, bioactive compounds have displayed hypouricemic properties also by interfering with the activity of the major urate transporters. Dietary flavonoids, especially fisetin and quercetin, contained in rooibos tea leaves have been shown to inhibit in vitro the activity of the urate transporter 1 (URAT1), an important renal urate reabsorber [34]. This biological action is combined with the activity against XOD described above, thus highlighting an interesting dual inhibitory activity for these compounds. They decrease UA production and inhibit transporters involved in urate reabsorption with an overall hypouricemic effect. This action, which was previously demonstrated for other molecules is important when considering the potential clinical application of bioactive compounds [35].

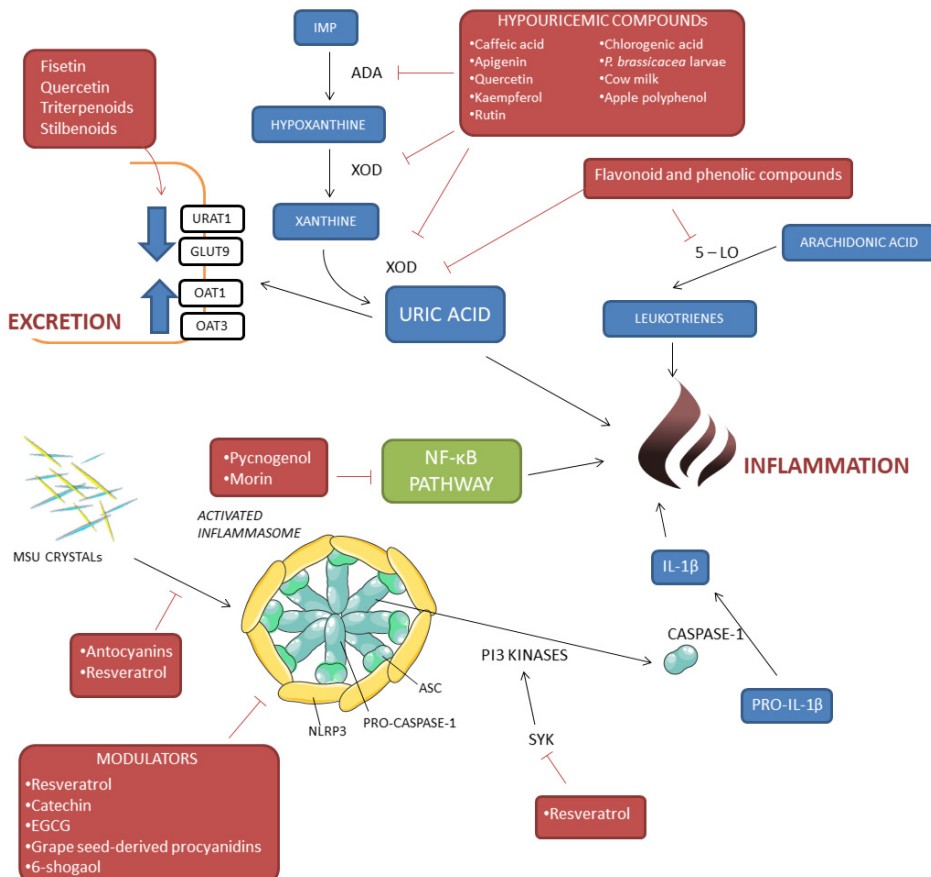

**Figure 2.** Molecular pathways affected by different bioactive compounds. Bioactive molecules can decrease uric acid levels acting on XOD and ADA enzymes, or hindering inflammation associated with MSU crystals deposits.

The promotion of UA excretion has been observed in MSU crystal-stimulated HK-2 cells treated with the triterpenoids Alstoscholarinoid A and B from Alstonia scholaris, which markedly downregulated GLUT9 and URAT1 expression [36].

### 3.2. Modulation of MSU Crystal-Induced Inflammation

As outlined above, the inflammatory mechanism triggered by MSU crystals involves a two-step process which finally leads to NLRP3 assembly, caspase-1 activation, and IL-1ß production. Bioactive compounds have been shown to exert their anti-inflammatory potential at different stages of these mechanisms (Figure 2) [37].

One of the bioactive compounds more extensively studied in MSU-crystal-induced inflammation is resveratrol [38], a plant-derived polyphenol mainly contained in grapes, red wine, and mulberry (Table 1). Resveratrol has been shown to inhibit the accumulation of acetylated α-tubulin and subsequent contact of mitochondria with the endoplasmic reticulum (ER), thus causing insufficient assembly of ASC and NLRP3 in macrophages [39]. Resveratrol also reduced intracellular pro-IL-1β synthesis in MSU crystal-stimulated monocytes through suppression of Syk phosphorylation [40]. Syk is a tyrosine kinase that activates the phosphatidylinositol 3-kinases (PI-3Ks) family involved in NLRP3 inflammasome-mediated caspase-1 activation [41].

**Table 1.** Principal induced animal models of disease in the study of bioactive compounds in gout and hyperuricemia.

| Animal Model | Induction | Features | Applications | Species | Reference |
|---|---|---|---|---|---|
| HYPERURICEMIA | Oral administration of a chemical inhibitor of uricase (potassium oxonate) | Increased serum uric acid concentrations. | Study of the influence on the metabolic pathway of uric acid, xanthine oxidase, urate reabsorbtion and excretion transporters | Mouse, rat | [42–54] |
| | Potassium oxonate and uric acid | Increased serum uric acid concentrations. | Study of the effects on urate reabsorbtion and excretion transporters | Mouse | [55] |
| | Potassium oxonate associated to purine rich diet | Increased serum uric acid concentrations | Analyze the association between dietary habits and hyperuricemia | Mouse Rat | [56–59] |
| | Oral gavage of hypoxanthine and oteracil potassium | Increased serum uric acid concentrations | Study of the influence on xanthine oxidase, urate reabsorbtion and excretion transporters | Mouse | [60] |
| GOUT | Intraperitoneal injection of MSU crystals | Acute inflammatory process with infiltrating immune cells consisting mainly of neutrophils | Study of therapeutic and prophylactic properties. Molecular pathways. | Mouse, rat | [61,62] |
| | Intrarticular injection of MSU crystals (knee, ankle) | Acute inflammatory arthritis with joint swelling, tissue inflammatory infiltrate, synovitis | Study of therapeutic and prophylactic properties. Influence on the development, and the different stages of disease. Molecular pathways. | Mouse, rat, rabbit | [54,61,63–68] |
| | Subcutaneous (paw) injection of MSU crystals | Acute inflammatory process with edema and tenosynovitis | Study of the effects on the inflammatory process. Molecular pathways. | Mouse, rat | [69–71] |

Table 1 shows the most widely used animal models of gout and hyperuricemia to assess the efficacy of bioactive compounds.

Resveratrol is one of the best-known activators of sirtuin1 [72]. It has been demonstrated that patients with gout have reduced levels of sirtuins with respect to healthy individuals and that resveratrol can restore these levels in PBMC collected from patients with gout and treated with MSU crystals. Through this mechanism, resveratrol has been shown to promote MSU crystal-induced autophagy and reduce IL-β release. Using the same experimental conditions, the transcriptional levels of NLRP3 and NF-kB p65 were shown to be regulated by the polyphenol [73].

The inhibition of IL-1β production is considered a therapeutic target in gout. Using the most traditional in vitro model of crystal-induced inflammation, some Authors observed that THP-1 cells pretreated with resveratrol or its natural precursor polydatin are unresponsive to MSU crystals stimulation. This effect was shown to be mediated, at least in part, by the capacity of resveratrol to inhibit the process of crystal phagocytosis [74]. Finally, resveratrol has been shown to influence the step of cell priming in MSU crystal-induced inflammation. In THP-1-derived macrophages resveratrol has been shown to prevent transforming growth factor-β activated kinase 1 (TAK1) activation and relieves MSU crystal-induced inflammation. It has been demonstrated that TAK1 activates the NF-κB signaling pathway, which in turn plays a critical role in the priming of the NLRP3 inflammasome in MSU crystal-induced inflammation [61].

Modulation of NLRP3 activation was observed also after treatment with catechins, phenolic compounds mostly found in cacao and tea (Table 1). The protective effects of catechin were highlighted by the inhibition of MSU-induced NLRP3 inflammasome activation and consequent IL-1β secretion and reduction of mitochondrial reactive oxygen species production and intracellular calcium levels. By contrast, the treatment with catechin up-regulated the levels of the cellular survival mitochondrial promoter Bcl-2 and restored MSU-induced mitochondrial transmembrane potential impairment, thus suggesting a modulatory effect on mitochondrial damage [75]. Furthermore, it has been demonstrated that green tea catechin, epigallocatechin-3-gallate (EGCG), suppressed NLRP3 inflammasome activation in MSU-challenged THP-1 monocytes [62]. These results confirm and extend those of a previous in vitro study, showing that EGCG can reduce the inflammatory response induced by calcium pyrophosphate crystals, a type of pathogenic crystal similar to MSU in triggering inflammation [76]. Grape seed-derived procyanidins also decreased MSU-induced activation of the NLRP3 inflammasome, IL-1β release, and ROS production in macrophages [77]. More recently, treatment with anthocyanins extracted from cherry demonstrated a reduction of IL-1β and ROS levels and crystal phagocytosis in THP-1 cultures stimulated by MSU crystals [78]. More recently, Erianin, a bibenzyl compound isolated from Dendrobium chrysotoxum Lindl, displayed a reduction of caspase-1 cleavage and IL-1β secretion MSU crystal-stimulated mouse bone marrow-derived macrophages. It has been supposed that Erianin directly interacted with NLRP3, leading to the inhibition of NLRP3 inflammasome assembly [79].

Additionally, 6-shogaol from ginger has evidenced an interesting inhibitory activity on NLRP3 inflammasome-mediated IL-1β production, but its effects were evaluated in cells previously treated with LPS [80].

Reduced production of other pro-inflammatory mediators, such as COX-2 and IL-8, induced by MSU crystal has been observed in different cell types cultured with pycnogenol and morin by inactivation of NF-kB signaling pathway (Table 1) [81,82]. Decreased levels of pro-inflammatory cytokines (IL-1β, IL-6, and TNF-α) have been determined also in MSU crystal-stimulated macrophages after treatment with polyphenols from purple potato leaves [83].

Finally, active ingredients from herbs, including berberine, astilbin, chlorogenic acid, caffeic acid, and ferulic acid, have demonstrated a protective effect on chatechon by reducing cell apoptosis [84]. Overall, the studies carried out in vitro showed a consistent effect of bioactive compounds in inhibiting or preventing the inflammatory process induced by MSU crystals. Nevertheless, the different concentrations of substances used in the various experiments make it a difficult task to translate the relevance of in vitro models to the in vivo situation.

## 4. Beneficial Properties of Bioactive Compounds in Animal Models of Gout

Animal models of gout and hyperuricemia are widely used to assess the efficacy of bioactive compounds in preclinical studies. However, mice do not develop gout or hyperuricemia spontaneously and the cut-off levels of pathological serum urate concentrations in the experimental model are disputed since rodents have lower levels of uricemia than humans.

The inactivation of the enzyme uricase with oral administration of potassium oxonate (PO) is a widespread model of hyperuricemia. After a week of intra-gastric treatment, a single dose of PO increases the levels of serum urate by 50–80% due to the impossibility of uricase to degrade urate in allantoin in liver tissues [42]. PO-induced hyperuricemia leads to enhanced activity of XOD and altered expression levels of renal urate transporter (URAT1) and organic anion transporter (OAT1) and is involved respectively in uric acid reabsorption and excretion [43]. Besides the effect on uricemia, this model affects systolic blood pressure and causes vascular damage in the endothelium and fibrosis in kidneys without the deposition of urate crystals [44].

Among bioactive compounds, stilbenes, including resveratrol, trans- 4-hydroxystilbene, pterostilbene, and polydatin, exert protective anti-inflammatory and antioxidant effects in hyperuricemic mice increasing OAT1 levels in kidneys of PO-induced hyperuricemic mice [45]. Mulberroside A, a resveratrol derivative, has been shown to exhibit uricosuric and nephroprotective properties in PO-treated mice, downregulating the expression of urate reabsorption transporters (URAT1, GLUT9), and increasing that of excretion transporters such as OAT1 [46].

Flavonoids such as quercetin and morin from Biota Orientalis lower serum uric acid levels in a dose-dependent manner inhibiting XOD in the same mouse model of disease [47].

The activity of XOD is inhibited also by green tea polyphenol administration which also enhanced the expression of organic anion transporter in the kidney [48].

A similar mechanism is shown by Baicalein, a flavonoid isolated from the roots of Scutellaria baicalensis Georgi [49]. Bioactive compounds including corilagin, gallic, and ellagic acid extracted from Longan seeds showed a uricosuric effect similar to that obtained with allopurinol, inhibiting XOD activity and modulating reabsorption transporters in kidney tissues [50].

The mouse model of PO-induced hyperuricemia has been used to test successfully the anti-oxidant effects of various polyphenols contained in Camellia japonica Bee Pollen.

These compounds have been shown to decrease the expression of URAT1, glucose transporter 9 (GLUT9) and increase that of OAT1 in kidney tissues, suppress the activation of inflammasome NLRP3, and influence gut microbiota composition [51].

A complementary model of hyperuricemia was developed in rats through a daily oral gavage of a mixture of PO and uric acid over 3 weeks. Using this model Lee and colleagues observed that resveratrol diminished uric acid serum levels by enhancing the expression of urate excretion transporter in kidneys, thus reducing its absorption at renal tubule levels [55]. In a similar model, in which hyperuricemia was established in mice by combined administration of PO and hypoxanthine, Sonneratia apetala (mangrove Apple) seed oil showed to downregulate XOD activity and to increase protein expressions of GLUT9, URAT1, and OAT1, thus showing the dual inhibitory activity described above [56]. The same activity has been reported for polysaccharides derived from Green Alga Ulva lactuca, Lycium barbarum L., and for Stevia residue extract [52,53,60].

Given the important association between hyperuricemia and the excessive consumption of meat, alcohol, and processed food, PO has been used along with the administration of high purine-content diets in laboratory animals [57]. Using this approach, some strains of probiotics, including *Lactobacillus Reuteri* TSR-332, *Lactobacillus Fermentum* TSF-331, and *Lacticaseibacillus paracasei*, have been shown to lower serum UA and relieve the damage at kidney tissue levels through inosine degradation and upregulation of urate transporters expression [58,59].

As hyperuricemic mice do not develop gout, several studies have been carried out injecting MSU crystals directly in the ankle, knee, or foot pad to resemble a gout acute attack which is mediated by NRLP3 activation. After injection, MSU crystals induce an acute inflammatory response with swelling and evaluable oedema at the site of injection that generally is self-limiting and resolves within 48–72 h from the induction [63,64].

Alternatively, MSU crystals are injected intraperitoneally to elicit strong recruitment and infiltration of leukocytes to reproduce a joint synovial cavity (Table 1).

Epigallocatechin gallate showed efficacy in inhibiting IL-1 β expression, blocking ROS production, and limiting leukocyte infiltration in the MSU peritonitis model [62]. Both models of MSU crystal-induced peritonitis and joint inflammation have been used to show the efficacy of resveratrol according to preventive and therapeutic treatments. This polyphenol demonstrated beneficial effects in reducing proinflammatory cytokines expression and limiting leukocyte infiltration through the inhibition of NRLP3 inflammasome and transcription factor NF-kB [54,61,69].

Highly oxygenated diterpenoids contained in diethanol extracts and various subfractions obtained from Clorodendrathus Spicatus showed a significative anti-gout efficacy in the model of gout as well as in PO-induced hyperuricemia. Extracts from Lychnophora trichocarpha and of Lychnophora Pinaster which contains flavonoids such as luteolin, apigenin, lupeol, lychnopholide, and eremantholide C, putin, quercetin, cinnamic and chlorogenic acid and stigmasterol showed both anti arthritic and anti hyperuricemic effects in PO-induced hyperuricemic mouse model and MSU foot pad experimental inflammation [65,70,71].

Recently, various polyphenols extracted from the edible fungus Phellinus Igniarius have been shown to improve both hyperuricemia and MSU crystal-induced arthritis in a rat model by inhibiting XOD activity and down-regulating the secretions of pro-inflammatory cytokines [66].

As far as pain in gout is concerned, a few studies have considered the effect of bioactive compounds on this important clinical feature. Using the model of MSU crystal-induced arthritis in the ankle, some authors demonstrated that eucalyptol attenuated joint pain via the downregulation of the expression of receptors involved in pain transmission (TRPV1) in dorsal root ganglion neurons of the ankle and fading IL-1 β activated nociceptors in peripheral sensory nerve system [67]. Similarly, stephalagine, an aporphine alkaloid extracted from Annona crassiflora fruit peel, has been shown to ameliorate inflammatory pain in mice acting on TRPA1 and TRPV1 channels modulation [68].

A summary of the beneficial effect exerted by bioactive compounds in different experimental model is shown in Table 2.

**Table 2.** Main bioactive compounds demonstrating beneficial effects in gout and hyperuricemia.

| Bioactive Compounds | Chemical Nature | Main Sources | Experimental Model | Main Benefits | References |
|---|---|---|---|---|---|
| Baicalein | Flavones | *Scutellaria* roots, thyme | PO-induced hyperuricemia in mice | ↓sUA | [49] |
| Caffeic acid | Hydroxycinnamic acid | Herbs, spices | XOD inhibition assay HUVECs | ↓XOD ↓intracellular ROS | [12,84] |
| Catechins | Flavonols | Cacao, tea | MSU crystal-activated THP-1 MSU crystals-induced peritonitis | ↓NLRP3 ↓IL-1ß ↓ROS ↓NFkB | [62,75] |
| Chlorogenic acid | Hydroxycinnamic acid | Dark chocolate, herbs, hard wheat | XOD inhibition assay HUVECs PO-induced hyperuricemia in mice | ↓XOD ↓cell apoptosis ↓ROS ↓sUA | [17,25,71,84] |
| Eucalyptol | Terpene | Eucalyptus oil, herbs, spices | MSU crystals-induced arthritis in mice | ↑TRPV-1 | [67] |
| Ferulic acid | Hydroxycinnamic acid | Cereals, fruits | HUVECs | ↓ROS | [84] |
| Morin | Flavonols | Strawberries, almonds, fig | LPS-stimulated RAW 264.7 cells MSU crystal-stimulated RAW 264.7 cells PO-induced hyperuricemia in mice | ↓XOD ↓ROS ↓NFkB ↓sUA | [31,46,82] |
| Pycnogenol | Procyanidins | Pine bark | MSU crystal-stimulated chondrocytes and synoviocytes | ↓COX2 ↓IL-8 ↓NFkB | [81] |
| Quercetin | Flavonols | Red onion, berries | HUVECs treated with high-glucose concentrations Kidney 293A cells PO-induced hyperuricemia in mice | ↓XOD, ↓ADA ↓URAT 1 ↓sUA | [13,14,34,47,71] |
| 6-Shogoal | Monomethoxybenzene | Dried ginger | LPS-stimulated THP-1 cells | ↓IL-1ß | [80] |
| Resveratrol | Stilbenes | Grapes, red wine | MSU crystal-stimulated mouse primary macrophages MSU crystal-stimulated human primary monocytes MSU crystal-stimulated PBMCs MSU-crystal-stimulated THP-1 MSU crystal-induced murine peritonitis PO-induced hyperuricemia in mice | ↓NLRP3 ↓IL-1ß ↓Syk ↓NFkB ↑Sirtuin ↓autophagy ↓TAK ↓Phago ↓sUA | [39,40,45,55,61,73,74] |

COX, cyclooxygenase; IL, interleukin; NFkB, nuclear factor kB; NLRP3, inflammasome; phago, phagocytosis; ROS, reactive oxygen species; sUA, serum uric acid; SYK, spleen tyrosine kinase; TAK, transforming growth factor-β-activated kinase; TRPV, transient receptor potential cation channel subfamily V; ↑, increase; ↓, reduction.

## 5. Clinical Studies on Bioactive Compound in Hyperuricemia and Gout

A few human studies have been carried out to determine the anti-gout activity of bioactive compounds as compared to in vitro and in vivo studies, and most of them focused on the reduction of uric acid levels. Table 3 shows the compounds used in clinical studies to target hyperuricemia and gout.

The uricosuric effect of Roselle (Hibiscus sabdariffa), a genus of the Malvaceae family rich in flavonoids and phenolic acid, was investigated in subjects with and without a history of renal stones. Both groups showed an increased uric acid excretion and clearance after the intake of a cup of tea made from dry Roselle calyces twice daily for 15 days, thus suggesting a potential beneficial effect in urate serum levels in gouty patients [85].

A reduction in plasma uric acid concentration was observed in an RCT considering pre-hyperuricemic males after daily oral supplementation for 4 weeks with quercetin, a flavonoid found mainly in onions, tea, and apples [86]. From the same family, catechins from green tea have been shown to enhance uric acid excretion after a short time from alcohol ingestion [87].

A decrease in serum uric acid levels and an increase in its clearance were found in subjects with mild hyperuricemia drinking for 12 weeks a fermented barley extract [88].

Conversely, healthy volunteers daily treated with a phenolic substance contained in lychee fruit, oligonol, evidenced significantly decreased 1-h uric acid excretion and fractional uric acid clearance. This was accompanied by a decreased serum concentration of uric acid, thus indicating that the effect of oligonol may be associated with XOD inhibition [89].

XOD activity inhibition has been demonstrated also for apple polyphenols in an RCT involving overweight volunteers with suboptimal values of fasting plasma glucose. In this study, the intake of apple polyphenols for 8 weeks resulted in an improvement in serum uric acid and fasting plasma glucose levels when compared with baseline and the placebo group [33].

Recently, an increasing number of studies have been conducted to determine the potential anti-gout efficacy of cherries and cherry products. In this context, a first case report in 1950 demonstrated that the uric acid content of the blood was reduced to normal by the daily consumption of about 1/2 Ib. fresh or canned cherries in gouty patients [90]. More recent studies provide evidence that consumption of cherries or cherry juice increased urinary urate excretion and reduced levels of serum urate and C-reactive protein in healthy and overweight adults, thus revealing not only a hypouricemic action but also an anti-inflammatory activity [91–94]. The effect of cherries consumption has been also investigated with regard to gout flares. Of note, an observational case-crossover study conducted in 633 gout patients, showed that cherry intake over a 2-day period was associated with a 35% lower risk of acute attacks, reaching 75% in combination with the use of allopurinol [95]. Using a different approach, Schlesinger et al. in a prospective study of patients ingesting cherry juice concentrate daily for $\geq$4 months reported a $\geq$50% reduction in gout attacks in approximately half of the patients. Interestingly, 36% of patients that not received urate-lowering therapy did not manifest acute attacks after 4–6 months of assuming cherry juice [96]. Improvements in gout flares after intake of cherry extract were confirmed in a pilot internet RCT involving 84 patients with gout followed for 9 months [97].

**Table 3.** Clinical studies carried out on bioactive compounds in gout and hyperuricemia.

| Bioactive Compounds or Plant | Target | Study | Cases, N. | Doses | Intervention Period | Main Results | Reference |
|---|---|---|---|---|---|---|---|
| Hibiscus sabdariffa | Hyperuricemia | nRCT | 18(9t + 9c) | 1.5 g/bid | 2 weeks | ↑UA excretion | [85] |
| Quercetin | Hyperuricemia | R-Crossover | 22(14t + 9c) | 0.5 g/d | 4 weeks | ↓sUA | [45] |
| Catechins | Healthy | R-Crossover | 10t | 617 mg | 5 h | ↑UA excretion | [87] |
| Barley extract | Hyperuricemia | RCT | 111(56t + 55c) | 2 g/d | 12 weeks | ↓sUA,↑UA excretion | [88] |
| Oligonol | Healthy | nRCT | 6t | 0.6 g | 1 h | ↓UA clearance, ↓XOD activity | [89] |
| Apple extract | Overweight | RCT | 62(31t + 31c) | 0.3 g/d | 8 weeks | ↓sUA,↓ER | [33] |
| Cherry concentrate | Healthy | R-Crossover | 60t | 30–60 mL/bid | 48 h | ↓sUA,↑UA excretion | [91] |
| Cherries | Healthy | nRCT | 10t | 280 g | 5 h | ↓sUA,↑UA excretion | [92] |
| Cherry juice | Overweight, obese | R-Crossover | 26t | 240 mL/d | 4 weeks | ↓sUA | [93] |
| Cherries | Gout | Case-crossover | 633 | Not reported | 1 year | ↓Gout flares | [95] |
| Cherry juice | Gout | RCT | Not reported | Not reported | 4–6 months | ↓Gout flares | [96] |
| Cherry extract | Gout | RCT | 84(41t + 43c) | 3.6 g/d | 9 months | ↓Gout flares | [97] |
| Polyherbal preparation | Gout | nRCT | 27t | 4 tablets/d | 45 days | ↓sUA | [98] |
| Cherry juice | Gout | RCT/protocol | 120 | 30 mL/d | 12 months | Outcomes: Gout flares, sUA, UA excretion | [99] |
| Cherry concentrate | Gout | RCT | 50(25t + 25c) | 7.5, 15, 22.5, 30mL/bid | 4 weeks | ↔sUA,↓gout flares | [100] |
| Probiotic/ *L.gasseri* PA3 | Gout | RCT | 25(13t + 12c) | 100 g/bid | 8 weeks | ↓sUA levels | [101] |
| Probiotic/ *L.gasseri* PA3 | Hyperuricemia | RCT | 60(20t + 20t + 20c) | 3–30 million CFU/d | 8 weeks | ↓sUA levels | [102] |

C, controls; bid, bis in die; d, daily; ER, endothelial reactivity; RCT, randomized controlled trial; nRCT, non-randomized non-controlled trial; R-Crossover, randomized crossover; t, treated; sUA, serum uric acid; XOD, xanthine oxidase. ↑, increase; ↓, reduction; ↔, no effect.

The beneficial effects of bioactive compounds from cherry were also observed when their assumption is combined with other phytochemicals from plants, as well as micronutrients, such as folic acid and potassium citrate. Indeed, in a single-arm, open-label pilot study in patients with gout that daily consumed orally a mixture of extract from fruit, including cherries, reduction of blood uric acid levels and improvement in pain and frequency of joint swelling were reported after 45 days of treatment [98]. Given the high interest in this field and the lack of an adequate control group in the studies conducted so far, as well as the limited sample size, a protocol for an RCT to evaluate the effect of tart cherry juice on the risk of gout attacks has been proposed in 2020. The primary outcome of this protocol is to examine the effectiveness of tart cherry juice to reduce the risk of recurrent gout, but it investigates mechanisms that may be involved [99].

In contrast to the studies described above, an RCT conducted on 50 patients with gout concludes that cherry concentrate does not have any effect on serum urate levels over a period of 8 weeks. In that study, the authors state that a possible effect on gout flares over a longer time period is not likely to be mediated by serum urate reduction [100].

Finally, two double-blind RCT studies investigated the effect of probiotics on gout and hyperuricemia. In the first study, patients with gout reported a decrease in serum uric acid levels after the intake of yogurt containing Lactobacillus gasseri PA-3 for 8 weeks [101]. A reduction of uric acid has been demonstrated in a cohort of patients with hyperuricemia receiving two doses of the same probiotic compared to a placebo [102].

For the beneficial effects listed above, bioactive compounds can have potential complementary benefits to current drugs used in gout and hyperuricemia. However, some interactions need also to be considered. Because of common metabolic pathways, allopurinol, a xanthine oxidase inhibitor, can affect theophylline metabolism and clearance and cause potentially harmful effects. Herbs containing theophylline might be therefore used with attention in those patients taking allopurinol for long-term therapy [103].

On the other hand, theophylline has been shown to inhibit allopurinol transport provoking a paradoxical attack of gout [104].

Colchicine causes reversible malabsorption in the gastrointestinal tract by disturbing epithelial cell function and inhibiting cell proliferation. This drug has been reported to reduce the absorption of beta-carotene which largely takes place in the gastrointestinal mucosa [105].

## 6. Conclusions

Gout is a common inflammatory condition that can lead to chronic deposits of pathogenic crystals in soft tissues with consequent disability. Currently, pharmacological therapies focus on preventing the disease and the development of acute flares through hypouricemic drugs, such as allopurinol and febuxostat which control serum uric acid levels. By contrast, inflammatory episodes are treated with colchicine, non-steroidal anti-inflammatory drugs, intra-articular injections of corticosteroids, and when necessary, biological drugs [106].

Although the management of gout can be properly achieved by a rheumatologist and general practitioners, it needs to be tailored for each patient according to current medications and comorbidities, and, more importantly, according to common adverse events associated with these therapies.

As outlined in this review, a growing body of evidence indicates that bioactive compounds may offer beneficial effects in patients with gout and hyperuricemia. Although it is difficult to extrapolate information on treatment and effect in humans (Table 4), the valuable effects obtained by in vitro and in vivo studies demonstrate a double interesting role in reducing both the inflammatory process caused by MSU crystals and the metabolic enzymes involved in uric acid production. More importantly, the preventive anti-inflammatory effect of these compounds pointed out in the experimental studies and more recently in a similar model of disease [107], emphasizes their role as prevention strategies for the patients.



**Table 4.** Evidence supporting the potential activity of bioactive compounds in humans.

| Classes | Bioactive Compounds | Animal Model | Outcomes | Results | Evidence in Humans |
|---------|---------------------|--------------|----------|---------|--------------------|
| Polyphenols [45,46] | Resveratrol, Mulberroside A | PO-induced hyperuricemia | Expression of urate excretion and reabsorbtion transporters | Increase (OAT1), decrease (URAT1, GLUT9) | Potential benefit on sUA levels through increase excretion [68,86,90,91] |
| Lactobacilli [58,59] | Lactobacillus Reuteri, Lactobacillus Paracasei | PO-induced hyperuricemia + high purine diet | Serum UA, Expression of urate transporters | Increase (OAT1, OAT3), decrease (URAT1) | Potential benefit on sUA levels [100,101] |
| Polyphenols [66,70,71] | Lychnophora Trichocarpha and Pinaster Extract, Fungus Phellinus Igniarius | MSU crystals-induced arthritis | Swelling, IL-1ß, ROS, XOD activity | Decrease | No evidence |
| Alcaloids, terpenes [67,68] | Eucalyptol, Stephalagine | MSU crystals-induced arthritis | Pain | Down-regulation TRPV1, TRPA1 | No evidence |

Subsequently, the intake of supplements or foods rich in bioactive compounds should be essential and recommended for patients with gout and those at higher risk of disease development.

Barley extract's main components are secondary metabolites of Aspergillus species and other antioxidants. Apple extract's main components are glycosylated phloritzin, chlorogenic acid, and quercetin. Cherry concentrate main component: cyanidin-3-O-glucoside rutinoside. Cherry juice's main components are anthocyanins. Polyherbal preparation ingredients: P. cerasus fruit extract (100 mg); A. graveolens seed extract (300 mg); H. procumbens tuber extract (150 mg); potassium citrate (50 mg); ascorbic acid (50 mg) and folic acid (90 μg).

**Author Contributions:** Conceptualization, A.S., R.L., R.R., P.S. (Paolo Spinella), P.S. (Paolo Sfriso), P.G. and F.O.; methodology, A.S. and F.O.; software, P.G.; validation, R.R., P.S. (Paolo Spinella) and P.S. (Paolo Sfriso); formal analysis, A.S., P.G. and F.O.; investigation, A.S., R.L. and F.O.; resources, A.S. and F.O.; data curation, F.O.; writing—original draft preparation, A.S., R.L. and F.O.; writing—review and editing, A.S., P.G. and F.O.; supervision, R.R., P.S. (Paolo Spinella) and P.S. (Paolo Sfriso); project administration, F.O. All authors have read and agreed to the published version of the manuscript.

**Funding:** This research received no external funding.

**Institutional Review Board Statement:** Not applicable.

**Informed Consent Statement:** Not applicable.

**Data Availability Statement:** Not applicable.

**Conflicts of Interest:** The authors declare no conflict of interest.

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
