# Peer review of "Anti-Inflammatory and Hypouricemic Effect of Bioactive Compounds: Molecular Evidence and Potential Application in the Management of Gout"

_cimb, doi:10.3390/cimb44110352_

Round 1
Reviewer 1 Report
Manuscript title: The beneficial role of bioactive compounds in gout: a comprehensive review
In the study, the authors have made overall review works in the investigation of the beneficial role of bioactive compounds in hyperuricemia, gout development and inflammatory pathways of gout. The reviewed references are sound and plainly exposed, explanations of conditions leading to hyperuricemia and gout in humans are adequate and the molecular pathways in the actions of bioactive compounds on the disease are clearly shown and discussed. However, the presentation of this study in the Tables 1 and 2 might be strengthened by adding the cited references in the table for reading as same as the Table 3.
The manuscript title: The beneficial role of bioactive compounds in gout: a comprehensive review.
Q. The period should be not presented in the title.
Author Response
Reviewer 1
- The presentation of this study in the Tables 1 and 2 might be strengthened by adding the cited references in the table for reading as same as the Table 3.
As suggested by the reviewer, references are now reported in the tables.
- The period should be not presented in the title.
The period has been removed from the title.
Reviewer 2 Report
This manuscript (MS) written by Anna Scanu et al. is a narrative review for providing an overview of the beneficial role of bioactive compounds in hyperuricemia, gout development and inflammatory pathways of the disease. The authors classified such bioactive compounds into two groups that influence on serum urate and inflammatory processes. Although the current title is “The beneficial role of bioactive compounds in gout: a comprehensive review”, “Anti-inflammatory and hypouricemic effect of bioactive compounds: molecular evidence and potential application in the management of gout” (in the footnote) may be better, given little evidence for anti-gout effects of such substances in humans. Given some concerns described below, descriptions in this MS should be carefully improved, which can not allow the Reviewer to recommend the publication of this MS in Current Issues in Molecular Biology at the current form. Also, the Reviewer reserves other comments at this moment.
Major comments:
Pathophysiological descriptions about gout and hyperuricemia should be correctly. For instance, hyperuricemia is defined as serum urate > 7 mg/dL, etc. In clinical situations, inhibitors for uric acid production (XOR inhibitors) and/or uricosuric agents are used for serum urate-lowering. Thus, bioactive compounds (which are not drugs) the authors focused on should be addressed from those two points of view. However, the authors only featured XOR inhibitory activity. Moreover, given that recent researches reported dual inhibitory activities on XOR and renal urate transporter, such dual activity will also be an important topic. Thus, the former part of this MS including Figures 1 and 2 should be reconstituted.
Evidence levels supporting the potential activity of featured compounds are not clear at this current form of MS. In other words, the authors should specify what is confirmed for their featured bioactive compounds: in vitro inhibitory activities (kinetics parameters and experimental systems), in vivo effects in animal models (information on treatment, experimental model, effects, extrapolability to humans), clinical benefits in human studies (outcome, relevant in health care), etc. The Reviewer feels that the current form of MS is not enough for this point; such information should be schematically provided.
Tables 1 and 2 need appropriate References in all columns.
Author Response
Reviewer 2
- Although the current title is “The beneficial role of bioactive compounds in gout: a comprehensive review”, “Anti-inflammatory and hypouricemic effect of bioactive compounds: molecular evidence and potential application in the management of gout” (in the footnote) may be better, given little evidence for anti-gout effects of such substances in humans.
The title has been replaced as suggested by the reviewer.
- Pathophysiological descriptions about gout and hyperuricemia should be correctly. For instance, hyperuricemia is defined as serum urate > 7 mg/dL, etc. In clinical situations, inhibitors for uric acid production (XOR inhibitors) and/or uricosuric agents are used for serum urate-lowering. Thus, bioactive compounds (which are not drugs) the authors focused on should be addressed from those two points of view. However, the authors only featured XOR inhibitory activity. Moreover, given that recent researches reported dual inhibitory activities on XOR and renal urate transporter, such dual activity will also be an important topic. Thus, the former part of this MS including Figures 1 and 2 should be reconstituted.
The definition of hyperuricemia has been corrected and the text was extended to include recent studies reporting dual inhibitory activities on xanthine oxidase and renal urate transporter. Figures 1 and 2 have been revised accordingly.
- Evidence levels supporting the potential activity of featured compounds are not clear at this current form of MS. In other words, the authors should specify what is confirmed for their featured bioactive compounds: in vitro inhibitory activities (kinetics parameters and experimental systems), in vivo effects in animal models (information on treatment, experimental model, effects, extrapolability to humans), clinical benefits in human studies (outcome, relevant in health care), etc. The Reviewer feels that the current form of MS is not enough for this point; such information should be schematically provided.
Table 1 has been implemented and Table 4 has been added to schematically provide additional information on the bioactive compounds considered in the manuscript, as suggested by the reviewer.
- Tables 1 and 2 need appropriate References in all columns.
References are now reported in the tables.
Round 2
Reviewer 2 Report
The authors made a revision for this narrative review; however, there are some issues should be addressed as follows.
Hyperuricemia should be defined correctly.
The authors described additional information of dual inhibitory activities based on a recent study (line 145-150); however, such a concept has been recognized in the related fielded from long ago. So, the authors should respect previous studies with appropriate references.
Regarding Table 1, main benefits of many lines should be corrected. Only in vitro experiments cannot demonstrate serum urate -lowering, which is in vivo action. To avoid misleading, in vivo effect on sUA and in vitro effects on protein function (potential for sUA-lowering) should be distinguished.
In line 294, modulating of the gene expression is not inhibition. So, dual inhibitory mechanism needs for more accurate explanation.
In Table 4, the left column addressed chemical nature, but the column title is Bioactive compounds. This should be corrected and the details of the compounds should be listed in an additional column.
Author Response
We thank the reviewer for her/his additional comments which further improve our manuscript.
Point-by-point response.
- We defined hyperuricemia and added a reference
- Thank you for this suggestion, we modified a sentence and added a reference.
- We modified table 1 according the reviewer’s suggestions
- We modified the sentence and we gave a more accurate explanation in the previous paragraph
- Table 4 has been modified
All new changes are tracked in green.